# Keratoconus Diagnosis: From Fundamentals to Artificial Intelligence: A Systematic Narrative Review

**DOI:** 10.3390/diagnostics13162715

**Published:** 2023-08-21

**Authors:** Sana Niazi, Marta Jiménez-García, Oliver Findl, Zisis Gatzioufas, Farideh Doroodgar, Mohammad Hasan Shahriari, Mohammad Ali Javadi

**Affiliations:** 1Translational Ophthalmology Research Center, Tehran University of Medical Sciences, Tehran P.O. Box 1336616351, Iran; sananiazi@sbmu.ac.ir; 2Department of Ophthalmology, Antwerp University Hospital (UZA), 2650 Edegem, Belgium; 3Department of Medicine and Health Sciences, University of Antwerp, 2000 Antwerp, Belgium; 4Department of Ophthalmology, Vienna Institute for Research in Ocular Surgery (VIROS), Hanusch Hospital, 1140 Vienna, Austria; 5Department of Ophthalmology, University Hospital Basel, 4031 Basel, Switzerland; zisisg@hotmail.com; 6Negah Aref Ophthalmic Research Center, Shahid Beheshti University of Medical Sciences, Tehran P.O. Box 1544914599, Iran; 7Department of Health Information Technology and Management, School of Allied Medical Sciences, Shahid Beheshti University of Medical Sciences, Tehran P.O. Box 1971653313, Iran; 8Ophthalmic Research Center, Labbafinezhad Hospital, Shahid Beheshti University of Medical Sciences, Tehran P.O. Box 19395-4741, Iran

**Keywords:** artificial intelligence, biomechanical phenomena, corneal topography, diagnosis, deep learning, keratoconus, machine learning, neural networks, computer, optical coherence

## Abstract

The remarkable recent advances in managing keratoconus, the most common corneal ectasia, encouraged researchers to conduct further studies on the disease. Despite the abundance of information about keratoconus, debates persist regarding the detection of mild cases. Early detection plays a crucial role in facilitating less invasive treatments. This review encompasses corneal data ranging from the basic sciences to the application of artificial intelligence in keratoconus patients. Diagnostic systems utilize automated decision trees, support vector machines, and various types of neural networks, incorporating input from various corneal imaging equipment. Although the integration of artificial intelligence techniques into corneal imaging devices may take time, their popularity in clinical practice is increasing. Most of the studies reviewed herein demonstrate a high discriminatory power between normal and keratoconus cases, with a relatively lower discriminatory power for subclinical keratoconus.

## 1. Introduction

Corneal disorders are the world’s second-leading cause of blindness [1,2]. Keratoconus (KC) is a progressive corneal ectasia characterized by a thinning and protrusion of the cornea. The development of KC is influenced by both genetic and environmental factors, with environmental factors such as eye rubbing and nocturnal ocular compression appearing to play a more significant role [3,4,5]. The prevalence of KC varies across different regions, ranging from 1 in 50 individuals in Central India to 1 in 2000 individuals in the United States [2,6]. In spite of therapeutic advances including corneal collagen crosslinking and intracorneal ring segments, patients with KC are still an important group requiring corneal transplantation worldwide, and young adults and children are most affected by the condition [2,7,8]. Moreover, a lack of consensus on how to define suspect, subclinical, and forme fruste keratoconus (FFKC) persists [9]. Finding an appropriate level of sensitivity is essential to screen mild or subclinical keratoconus (SKC) to prevent iatrogenic keratectasia after laser refractive surgery [10,11].

Various methods were described for detecting keratoconus (KC) (Figure 1), primarily utilizing corneal topographers or tomographers. However, as mentioned, KC is a multifactorial condition involving genetic factors [12,13,14], environmental factors such as atopy [15], and repetitive mechanical corneal trauma [12] in its development and progression [16]. Due to the complex nature of KC, there is potential benefit in utilizing artificial intelligence (AI) approaches, including corneal biomechanical information, which already showed promise in forecasting the progression of keratoconus [17,18,19,20].

This review provides a state-of-the-art assessment of the indices (both traditional and based on AI) used in early keratoconus detection.

Since several previous studies reported on the sensitivity, specificity, and accuracy of KC detection and progression (Figure 2), we aim to establish a multidimensional comparative network to assess the best KC diagnosis and management approach.

We discuss the Placido disk imaging topographers, including EyeSys VK (EyeSys Vision, Houston, TX, USA); tomography devices, including the Orbscan II (Bausch & Lomb, Laval, QC, Canada), the Pentacam (Oculus, Wetzlar, Germany), the Galilei (Ziemer Ophthal-mic Systems, Port, Switzerland), and anterior segment OCT: RT-Vue-100 (Optovue, Fremont, CA, USA), with their relevant indexes (Figure 3). Corvis ST (CST; Oculus, Wetzlar, Germany) as a tomography and topography, combined with a biomechanical analyzer—as mentioned by Vinciguerra et al.—is evaluated [42]. In addition, the Sirius, epithelial thickness mapping, corneal biomechanics, and polarization-sensitive optical coherence tomography are also discussed.

### 1.1. Placido Disk-Based Corneal Topography

Placido disk-based corneal topography integrates the information received from the anterior corneal surface into a high-resolution color-coded topographic map of the cornea and calculates corneal curvature by the size and size distortion of the mires [43]. Computerized Placido ring videokeratoscope devices, such as the Topographic Modelling System (TMS–1; Computed Anatomy Inc., Ney York, NY, USA), map the anterior corneal surface using a digital camera that captures images reflected from concentric light rings [43,44].


**
KISA% index
**


Rabinowitz et al. developed the first method to distinguish KC from the normal cornea via Placido disk-based corneal topography [45], using the KISA% index. It is based on the keratometry (K) value, inferior–superior (I–S) value, relative skewing of the steepest radial axes (SRAX), and the keratometric astigmatism (AST) and is calculated as follows [46]:(1)$$KISA\%=\frac{\left(K\right)\times (I-S)\times (AST)\times (SRAX)\times 100}{300}$$

KISA% > 100% indicates keratoconus, KISA% = 60–100% indicates keratoconus suspect, and KISA% < 60% is considered normal [21].

However, steepening in the lower cornea does not occur in all patients with KC. In some patients, the difference between the nasal and temporal corneas may indicate keratoconus, in which case this index may be normal [45].


**
KC prediction index (KPI) and KC severity index (KSI)
**


KPI combines eight topography-based values—Flat Simulated K1 (Sim K1), Steep Simulated K2 (K2 Sim), Surface Asymmetry Index (SAI), Differential Sector Index (DSI), Opposite Sector Index (OSI), Center/Surround Index (CSI), Irregular Astigmatism Index (IAI), and the Analyzed Area (AA: the ratio of the interpolated data area to the area circumscribed by the outermost peripheral ring)—and can differentiate between normal corneas, regular astigmatism, peripheral, or central KC (Figure 4). A multivariate analysis obtained a cut-off of 0.23 for KPI, and values above this cut-off are suggestive of KC [22].

The KC severity index (KSI) in TMS–1 and OPD–Scan (Nidek Inc., Tokyo, Japan) focused on KC severity, distinguishing between healthy, suspect, and keratoconic corneas. The algorithm was based on a neural network with 10 topographic indices as inputs. KSI of 15% indicates normal, 15–30% suspect keratoconus, and >30% subclinical [23].


**
AI in Placido disk-based corneal topography
**


Although several studies reported good sensitivity and specificity for distinguishing clinical KC from healthy cases using Placido with or without additional diagnostic modality, it is still challenging to detect subclinical or forme fruste cases and stratify them [47,48,49].

In 1995, Maeda, Klyce, and Smolek [24] described applying AI and NNTs in the detection and evaluation of seven videokeratography categories: normal, astigmatism (with-the-rule), keratoconus (in three stages), post-photorefractive keratectomy, and post-keratoplasty. As a result, for each category, the accuracy and specificity were both more than 90%, although the sensitivity varied from 44% to 100%.

A retrospective analysis based on random forest on Zernike polynomials (obtained from an OCT integrated with Placido, the MS-39, CSO, Scandicci, Italy) achieved excellent area under the curve (AUC) accuracy and precision for healthy, KC, and very asymmetric ectasia (VAE) cases. However, the recall was 71.5 for VAE, lower than the recall for the KC and the healthy groups [48].

AI was also used to diagnose and grade keratoconus patients using deep learning on color-coded maps obtained from Placido disk-based corneal topography [49]. While the diagnosis of clinical KC had high accuracy, the accuracy was lower (86.8%) in subclinical keratoconus [49].

Machine learning algorithms such as naive Bayes classifiers fed by Placido disc-based topographer corneal indices were successfully used in KC diagnosis as well. They inherit all the advantages of the primary indices described above, while providing additional robustness to noisy or incomplete data [25,50].

### 1.2. Orbscan

Placido-based topography cannot provide any information about the posterior surface of the cornea and only covers about 60% of the corneal surface [26]. To overcome these limitations, alternative devices were developed, including the Orbscan IIz (Bausch and Lomb, Rochester, NY, USA) [51] (Figure 3 and Figure 4), which includes the SCORE.

As Rainer et al. mentioned, measurement of the corneal thickness has become an essential part of corneal refractive procedures not only to avoid complications such as postoperative keratectasia after LASIK but also to be applied in keratoconus management [52].

The SCORE is an AI-based analyzer that assists in the early diagnosis and follow-up of KC. The optimum cut-off, corresponding to zero, was established using a Receiver Operating Characteristics (ROC) curve, where positive and negative values represent a cornea with and without keratoconus, respectively [6]. The radar map represents six of the most discriminant topographic indices to calculate the SCORE.

Pachymetry of the thinnest point;Maximum posterior elevation in the central 3 mm;Irregularity in the central 3 mm;Vertical decentration of the thinnest point;Difference between mean central pachymetry (central zone 2 mm in diameter) and the pachymetry of the thinnest point;I–S value [6].


**
AI-based Orbscan analysis 
**


As Saad and Gatinel [10] explained using machine learning, e.g., discriminant analysis on Orbscan parameters, the accuracy of detecting suspect keratoconus increases [53]. Several studies compared several machine learning approaches in terms of accuracy [2,26,27,54,55,56,57,58]. A recent study used a Convolutional Neural Network in KC diagnosis including “elevation against the anterior best-fit sphere” (BFS), “elevation against the posterior BFS”, axial anterior curvature, and pachymetry maps [56].

Although a super vector machine (SVM), with a 95% accuracy rate in classifying normal and keratoconus, outperformed four other classifiers (k-nearest neighbors, decision trees, radial basis function, and multilayer perceptron), all classifiers had good precision when using all descriptors in a recent study [57].

### 1.3. Pentacam Comprehensive Eye Scanner

Recent advancement in technology enabled assessing the posterior cornea and obtaining pachymetry maps using three-dimensional tomographic images, such as rotational Scheimpflug imaging or optical coherence tomography [59]. The Pentacam comprehensive eye scanner (Oculus Optikgerate GmbH, Wetzlar, Germany) is one of the most popular devices [60]. Other devices that use the same principle include the TMS–5 (Tomey Corp., Nagoya, Japan), Sirius (CS), Costruzione Strumenti Oftalmici, Florance, Italy), and Galilei (Ziemer, Port, Switzerland). 

The multiple calculations of the Pentacam make it an applicable tool in the diagnosis of glaucoma, power of IOL, cataracts, corneal ectasia, and KC. When compared with other tomographic devices, like the Galilei and Orbscan II, the Pentacam exhibits excellent intra–device precision but inconsistent inter-device repeatability [61]. Some also suggested the decreased reliability of the Pentacam in the periphery, but it is still superior to previous technology, such as the Placido disk, and can diagnose diseases of the periphery with acceptable accuracy [62].

As the Pentacam measures and calculates several parameters, including topographic, tomographic, and pachymetric parameters, it provides large data that can be used for several purposes (Figure 3 and Figure 4).


**
Pentacam Topographic indices
**


The topographic indices include the central keratoconus index (CKI), the keratoconus index (KI), the index of height asymmetry (IHA), the index of height decentration (IHD), the index of surface variance (ISV), the index of vertical asymmetry (IVA), minimal sagittal curvature (Rmin), and posterior elevation (PE) [63]. As the elevation data correspond with the Amsler–Krumeich severity index [64], ophthalmologists consider PE data to be a very sensitive and precise diagnostic index for the identification of subtle changes in KC [65]. However, its value in SKC is not determined yet, and there are several controversies even for KC detection [66].

Similar to PE, most of the topographic indices, including anterior elevation, CKI (the ratio of the mean radius of curvature values in a peripheral Placido ring to the central ring), KI (the ratio of mean radius of curvature in upper and lower segments of the cornea), and IHA, have high diagnostic accuracy in the discrimination of clinical KC from normal eyes [67]. However, they fail to discriminate cases with SKC from normal eyes and were, therefore, suggested to be used with caution for patients suspected of SKC or in combination with other parameters [68].

A comparison between the Pentacam indices showed the superiority of KI compared to CKI [65], while others suggested less robust diagnostic accuracy for KI compared to other Pentacam parameters [69].

Rmin, which corresponds to the minimum sagittal curvature, is another index, which failed as an exclusive index to be used for diagnosis of KC or SKC [70].

However, the other topographic Pentacam indices showed promising results for both KC and SKC. ISV showed promising results for patients with KC and SKC [65] as well as the identification of disease progression [71]. Also, IHD, which calculates the degree of vertical centration from a Fourier analysis, was able to discriminate SKC and unilateral KC [70,72], but some questioned its reliability in SKC [73]; therefore, its validation requires further studies. IVA (which represents the curvature symmetry data with respect to the horizontal meridian) is also a valid index for both clinical and subclinical KC [73,74,75] and was suggested as the second-most-accurate Pentacam index for the diagnosis of KC [65]. Some also suggested a higher diagnostic accuracy for IVA in discrimination of unilateral KC in the normal vs. fellow eye, compared to some of the pachymetric indices [70].


**
Pachymetric indices
**


The pachymetric indices of the Pentacam, which include Ambrósio relational thickness (ART; min, max, and average, see Figure 4), central corneal thickness (CCT), pachymetric progression indices (PPI; min, max, and average), the Pentacam random forest index (PRFI), and thinnest corneal thickness (TCT), determine the severity of KC, most of which are considered valid indices for clinical and subclinical KC [28].

The Pentacam data on the anterior (A) and posterior (B) curvature within the 3 mm zone surrounding the thinnest point of the cornea, thinnest pachymetry (C), and best corrected distance visual acuity (D) are incorporated in a novel classification system for KC staging, the ABCD classification [76], which overcomes the limitations of the Amsler–Krumeich classification and KSI by considering the corneal thickness at the thinnest point as opposed to a central apical reading, which may be significantly altered in KC. Moreover, it also applicable for the identification of unilateral KC [76].

PPI, which calculates the change in corneal thickness over 360 degrees of the cornea, with the mean average reported at 0.13, mean max at 0.85, and mean min at 0.58 [77], is considered a better index than single-point measurements for the diagnosis of KC as well as SKC [78], although a few showed an area under the curve (AUC) of <0.90 for PPI [70,72].

ART, the ratio between the thinnest point and PPI, is a novel parameter for the diagnosis of KC [78,79,80] with the highest diagnostic accuracy at a cut-off value of 300–400 µm [81], while the results of studies are controversial concerning the diagnostic value of ART in cases of SKC [70,82,83].

The Belin–Ambrósio’s enhanced ectasia display total deviation (BAD-D) value is a multivariate index that provides a global characterization of the cornea by integrating pachymetric and elevation data, and the final overall map reading is reported as D, which contains 6 indices. The software presents suspect KC in yellow (values of 1.6–2.6) and clinical KC in red (values > 2.6). BAD_D showed a high value for the diagnosis of clinical KC and SKC [74]; however, more studies are required for its validation in cases with unilateral KC [70].

PRFI was introduced as a sensitive tool for screening ectasia, though it misdiagnosed some of SKCs and is inferior to the BAD_D novel index [29].

Although some studies considered CCT [84] and TCT [78,85] as valid indices for screening KC, especially in cases with topographic asymmetry, there are many controversies about these two parameters, and several studies did not find CCT [86,87] and TCT [86,87] accurate in the diagnosis of keratoconus.

Zernike polynomial modeling could discriminate cases with SKC from normal cases with high accuracy, especially the Zernike fitting indices of the corneal posterior elevation (AUC: 0.951) [88].


**
Comparison between
**
**
 the diagnostic ability of Pentacam AI indices
**


Comparing the diagnostic ability of the topographic and tomographic indices of the Pentacam for differentiating KC from normal corneas, PRFI and IHD were demonstrated to be the best indices [89]. This is in line with the earlier suggestion by Lopes et al. of PRFI as an accurate index for the identification of patients at risk of ectasia with a high accuracy (AUC = 0.98) [29]. Similarly, Kovács et al. approached the VAE-NT detection problem using NNTs with 0.96, 92%, and 85% values—AUC, sensitivity, and specificity, respectively [30]. In another study, IHD was suggested as the Pentacam index with the highest diagnostic accuracy for KC (AUC = 0.979), followed by IHA (AUC = 0.884) [72]. Others also confirmed IHD as a more sensitive index than BAD-D for diagnosis of KC (AUC = 0.97 vs. 0.89) [30]. Nevertheless, others reported the Pentacam I–S to be the best diagnostic index for KC with an AUC of 0.99 [78]. The difference in the results might be due to differences in the KC stage among the studies. Researchers also reported that the posterior and anterior curvature-based indices (such as SIb and I–S) had a better diagnostic ability for diagnosis of KC, compared to pachymetric and elevation-based parameters [70].

PRFI (with AUC = 0.847, sensitivity = 71.7%, and specificity = 87.9%) and I–S (with AUC = 0.862, sensitivity = 80.1%, and specificity = 79.2%) were the best indices when distinguishing SKC from normal eyes [89]. Similar to these results, two other studies also reported I–S as the best diagnostic index of the Pentacam for SKC with an AUC of 0.799 and 0.840, respectively [70,90]. On the other hand, others reported other indices as the best Pentacam index for distinguishing SKC from normal eyes such as BAD_D, IVA, ISV, and fifth-order vertical coma aberration [74]. In VAE, PRFI showed a higher diagnostic accuracy than the I–S index [29,91]. KISA, suggested as an accurate index for the diagnosis of early KC and SKC [80], was not found accurate enough to detect SKC in another study [92]. It was suggested that the discrepancy in the results among studies may be related to different definitions of SKC [93].

Steinberg et al. described a classification and regression tree (CART) algorithm, used the Corvis parameters, and included suspected cases of keratoconus [94]. In vivo biomechanical analyses (CST) showed only marginal improvement in KC screening protocols.

Recently, Issarti et al. proposed computer-aided diagnosis (CAD) for suspect KC detection, which obtained >95% sensitivity and specificity for suspect keratoconus, outperforming BAD-D [31]. The same research group proposed LOGIK, which can be used to stratify the KC stage [95] but achieved lower sensitivity and specificity for SKC detection [32] than CAD while still outperforming BAD-D [32,95]. Neither CAD nor LOGIK are yet available in the Pentacam.

The Pentacam data were also used to predict KC progression [17] and the need for CXL.

### 1.4. Galilei Corneal Tomography

The Galilei (ZiemerOphthalmic Systems AG, Port, Switzerland) integrates the elevation data of the Scheimpflug technology with corneal curvature data from Placido disk topography.

The indices used in the Galilei system are as follows:


**
Asphericity Asymmetry Index (AAI)
**


AAI or the Kranemann–Arce index measures the asymmetry of asphericity of the corneal surface and showed excellent diagnostic accuracy (100% sensitivity and 99.5% specificity) for the diagnosis of clinical KC. However, a lower diagnostic accuracy was reported for posterior AAI in cases with SKC, but, still, it was the highest among all 55 Galilei parameters [33]. Others also reported an acceptable area under the curve (AUC) with different values as optimum cut-off values of posterior AAI for the diagnosis of SKC [34]. Nevertheless, anterior AAI could not discriminate SKC from normal eyes [96].


**
Center/Surround Index (CSI)
**


CSI quantifies the difference in area-corrected corneal power, calculated by the mean axial keratometric power between the central area of the cornea (3–mm diameter) and a surrounding annulus (diameter of 3–6 mm) [97]. CSI is a reliable index for diagnosis of KC, while low AUC values were reported for the discrimination of SKC [98,99].


**
Differential Sector Index (DSI)
**


DSI refers to the degree of corneal surface asymmetry, calculated based on the eight-sector pattern of the corneal surface. High values are observed in peripheral KC and low-to-moderate values in central KC. Similar to CSI, it is a reliable tool for the diagnosis of KC but not SKC [24].


**
Objective Scatter Index (OSI)
**


Similar to DSI, OSI is a valid index for frank KC but not SKC [24,99].


**
Higher Order Aberrations (HOAs)
**


HOAs are measured by wave front analysis and expressed by Zernike polynomials. When computed with the Galilei system based on the corneal elevation, less variability was reported compared to other systems, possibly because of the dual-channel system of the Galilei [100]. HOAs, especially coma, are also used for estimating KC progression and diagnosis of early changes in the corneal surface, applicable in cases suspected of SKC. Vertical coma is different between SKC and normal corneas and higher in progressive vs. non–progressive KC, which suggests that this index may be able to show early changes in the corneal surface; however, further studies calculating AUC are required [96,101].


**
Irregular Astigmatism Index (IAI)
**


IAI measures the average sum of the area-corrected variation in the axial keratometric power between central rings of any given meridian of the corneal surface, which showed excellent diagnostic accuracy for KC [24], while different AUCs were reported for SKC, although none above 0.90, which rejects the recommendation of this index as an independent index for SKC screening [98,99].


**
Inferior–Superior (I–S) value
**


I–S calculates the dioptric asymmetry between inferior and superior cornea; values ≥ 1.4 D suggest the possibility of KC [46], although it has a low sensitivity for SKC, even when calculated by the Galilei [98]. The high specificity (90.7%), suggested by Shetty and colleagues, recommends this index as appropriate for ruling in SKC [98].


**
Surface Asymmetry Index (SAI)
**


SAI calculates the difference in keratometric power of the cornea between opposite points among 128 meridians and is considered a valid stand-alone index for the diagnosis of KC but not SKC. The combination of SAI with the posterior best-fit sphere resulted in a high diagnostic accuracy in SKC (100% sensitivity and 91.3% specificity) [99].


**
Surface Regularity Index (SRI)
**


SRI shows the local irregularities by the sum of power variations among 256 semi-meridians on the corneal surface; a smooth corneal surface shows an SRI of zero, and values < 1.55 are accepted as normal. The SRI index has a high diagnostic accuracy for KC and is reported to have an AUC of 0.875 for cases with SKC, comparable to the BAD-D index of the Pentacam system [98]. However, more studies are required to ascertain the diagnostic value of SRI in SKC.


**
Total Corneal Power (TCP)
**


TCP, calculated by ray tracing, is the average of the corneal power at each point. The standard deviation of the corneal power (SDP) is also provided. TCP-steep and TCP-central showed an AUC > 0.9, while TCP-flat showed a lower AUC (0.79) for the diagnosis of KC [102]. For patients with SKC, TCP had an AUC of 0.887 and was superior to BAD_D [98,99,103].


**
Cone Location and Magnitude Index (CLMI)
**


CLMI, at first based solely on information from the anterior corneal surface, was proposed in 2008. The addition of posterior surface and corneal thickness information increased its diagnostic accuracy to >99% [104]. Although considered an excellent index for KC, it was shown to have inadequate diagnostic accuracy for SKC [98].


**
Keratoconus Prediction Index (KPI)
**


KPI is a multivariate topographic index that calculates the probability of KC based on analysis of anterior corneal surface, which includes simulated keratometry, DSI, OSI, CSI, SAI, IAI, and percentage area. It was shown to have excellent diagnostic accuracy for the differentiation of KC [98,99] but not for SKC [34].


**
Keratoconus Probability (Kprob)
**


Kprob uses a normative and keratoconic database for estimating the sensitivity and specificity of the reported KPI, which showed excellent AUC for KC (>0.99) but lower AUC (0.6) for SKC [98,99].


**
Percentage Probability of Keratoconus (PPK)
**


PPK refers to the optimal threshold for diagnosis of KC, calculated based on an equation using CLMI and axial data with excellent diagnostic accuracy for KC but not for SKC [98,103].

The Galilei indices, except for posterior AAI, showed a low diagnostic accuracy for SKC but excellent diagnostic accuracy for KC. It was, therefore, suggested to use a combination of the Galilei indices for SKC [103] (Figure 3 and Figure 4). 


**
Machine Learning-based analysis for Galilei
**


A recent study proposed an automated decision tree model that achieved high sensitivity and specificity (93.6% and 97.2%, respectively) to detect subclinical KC [33]. AAI and corneal volume were the most relevant parameters in the diagnosis of forme fruste KC [33]. However, another study concluded that the combination of the posterior best-fit sphere radius presented the highest prediction accuracy [99].

### 1.5. Sirius

The Sirius (Costruzione Strumenti Oftalmici, Florence, Italy) is a hybrid device that combines Placido and the Scheimpflug system [105]. Artificial intelligence based on neural networks displays critical information regarding the risk of post-surgery ectasia. The Sirius integrates a variety of tools for KC screening.


**
Symmetry Index front (SIf) and Symmetry Index back (SIb)
**


SIf demonstrates the symmetry index of the anterior curvature, and SIb demonstrates the symmetry index of the posterior curvature. The discrepancy in average size between the upper and lower hemispheres indicates the existence of keratoconus [106]. The inferior cornea is likely to have more curvature than the upper half in KC, but this difference is minimal in a healthy cornea. The highest specificity to detect KC and suspect KC compared to normal was reported for SIb, 99.9% and 84.7%, respectively [107].


**
Keratoconus Vertex front (KVf) and Keratoconus Vertex back (KVb)
**


KVf and KVb represent the highest elevation of the anterior and posterior corneal surface, respectively. The best aspherotoric surface is employed as a reference level. KVb demonstrated an outstanding diagnostic ability (AUC: 0.999) to identify KC from normal corneas in a study [89].


**
Baiocchi Calossi Versaci (BCVf/b; HOAs)
**


The BCV acronym derives from the initials of the three authors, Baiocchi, Calossi, and Versaci, who led the study. As KC is more common in the inferotemporal region of the cornea, it was hypothesized that this area has a higher impact on aberrations. The following HOA components were considered while calculating the BCV index:

Vertical trefoil $Z_{3}^{-3}$;Vertical coma $Z_{3}^{-1}$;Horizontal coma $Z_{3}^{+1}$;Primary spherical aberration $Z_{0}^{4}$;Second-order vertical coma $Z_{5}^{-1}$.

The BCV parameter is vectorially computed for the anterior and posterior corneal surfaces. In normal corneas, the values of the parameters are generally near to zero; even if the BCVf and BCVb have values larger than zero, the overall result is almost certainly close to zero [106]. A recent study [53] found that BCV as a combination index of high-order aberrations had perfect sensitivity and specificity for diagnosing KC, which is consistent with earlier studies [98,108] that demonstrated a much more pronounced progression in BCV in suspicious and keratoconus eyes compared to normal eyes. The predicted accuracy of BCVf and BCVb were pretty close (0.999 and 0.998, respectively), which may suggest that investigating both indices would be equally important [53].


**
Thinnest point of the cornea (ThkMin)
**


Corneal thinning is one of the most common keratoconus signs. This parameter is obtained from the thinnest corneal thickness in the 8 mm zone, which is used to diagnose keratoconus by comparing its changes with the total thickness of the cornea in suspicious eyes compared to normal eyes.

The Sirius can show not only the thinnest point but also several essential parameters in the anterior and posterior tangential maps, the anterior and posterior elevation maps, and the pachymetric map [107] such as:AKf-Apical Keratoscopy Front: the steepest point of the anterior corneal surface;Akb-Apical Keratoscopy Back: the steepest point of the posterior corneal surface;KVf-Keratoscopy Vertex Front: the highest point of ectasia on the anterior corneal surface;KVb-Keratoscopy Vertex Back: the highest point of ectasia on the posterior corneal surface.


**
Sirius and AI
**


Arbelaez et al. used a support vector machine (SVM) that achieved 93% true predictions when based solely on anterior surface data to classify normal, post-refractive surgery (abnormal), KC, and SKC corneas. However, including data from the posterior surface and from pachymetry had a positive impact, increasing accuracy especially in the detection of SKC [75]. EMKLAS, based on ordinal logistic regression, used demographic and tomographic data to detect mild and early KC and obtained acceptable accuracy in the mild KC group, whereas the accuracy was remarkably lower in the early KC subgroup [35]. Exploring additional ML techniques and larger datasets may increase the quality of the results.

### 1.6. Optical Coherence Tomography

One of the early alterations in KC is the lower basal epithelial density and degeneration of the basal epithelial layer [109]. Previous studies found that four OCT parameters best characterize keratoconic corneal thickness variations [109,110,111].

The minimal thickness of the cornea (Min);The minimal corneal thickness minus the highest corneal thickness (Min–Max);The typical variation between the superonasal and inferotemporal corneal thicknesses between rings of two and five diameters (SN-IT);The epithelial standard deviation (Std Dev) [36,111].

A recent study based on a logistic regression model aimed to classify corneas as normal (including contact lens warpage) or KC and obtained an accuracy of 100 ± 0% for normal and 99.0 ± 2.0% for KC (including all stages) [112]. When analyzing the KC subgroups, the accuracy for forme fruste KC dropped to 53%.

It should be noted that stromal thinning may be underestimated due to subsequent epithelial thickness alterations as a compensatory mechanism after stromal augmentation (Figure 5) [113].

In an ectatic cornea, the epithelium is typically thicker than in a normal cornea, even though it is thinner above the keratoconus protrusion to a degree that is significantly less than what was anticipated. This difference between normal eyes, patients with untreated keratoconus eyes, and patients with keratoconus eyes treated with CXL appears to be clinically significant, and epithelial pachymetry measurements using High-Frequency Ultrasound Biomicroscopy (HF UBM) may be used as a screening tool for ectasia-prone eyes [114].

Moreover, corneal epithelial thickness (ET) can be affected by a variety of different factors, including age, gender, dry eye, axial length, high myopia, contact lens use, laser refractive surgery, measurement area, and the presence of corneal ectatic disorders [115].

The accuracy of diagnosis is increased by recognizing the propensity toward AI, which goes above and beyond what is possible with plain quantitative indicators.


**
OCT and Artificial Intelligence
**


ML on OCT data was used not only for diagnosing and grading corneal ectasia [116] (Figure 3) but also for predicting the likelihood of future keratoplasty [117]. Different approaches were used, such as random forest analysis [118], neural networks [118], or deep convolutional neural network [119].

Yousefi et al. used an unsupervised machine learning algorithm including principal component analysis (PCA), manifold learning, and density-based clustering on OCT data in a stepwise process to minimize the error prediction rate. They found that the keratoconus stages and progression can be identified using unsupervised machine learning algorithms [116].

A deep convolutional neural network based on AS-OCT was able to identify diverse corneal diseases (including corneal epithelial defects, epithelium thickening, corneal thinning, etc.) and provided tissue stratification of the corneal epithelium and stroma [119].

### 1.7. Biomechanical Measurements

KC is characterized by the loss of stromal fibrils, a change in fibril orientation, the reduced cross-linking of collagen fibers, and the dysfunction of keratocytes, which decrease the mechanical stability of the cornea. As the corneal biomechanical changes may precede the tomographic changes, in vivo evaluation of corneal biomechanics was proposed for the early diagnosis of KC.

#### 1.7.1. Ocular Response Analyzer

The first device that became commercially available for the measurement of corneal biomechanical properties was the Ocular Response Analyzer (ORA; Reichert Ophthalmic Instruments, Buffalo, NY, USA), The ORA consists of a metered air-pulse emitter, an infrared emitter, and a collimation detector that temporarily indent the cornea and concurrently measure the infrared reflectance of the ocular response [120]. The two reflectance peaks produced, P1 (while the cornea is moving inwards) and P2 (while the cornea is moving outwards), indicate the points of the maximum planar surface area of the cornea within a 3 mm sampling zone. The outputs of the ORA include corneal hysteresis (CH; the difference between the air pulse pressures of the two points) and the corneal resistance factor (CRF; the difference between the air pulse pressures of the two points with a k factor of 0.7 (P1–kP2), considered to maximize the dependence of this parameter on the central corneal thickness. After multiple investigations, the overlap between KC and healthy corneas on the provided parameters remains an unsolved issue [120].


**
ORA and artificial intelligence
**


Some studies tried to evaluate ORA performance in distinguishing forme fruste keratoconus (FFKC) [121] and subclinical keratoconus (grades I and II keratoconus) [122] from normal corneas. A logistic regression model obtained higher AUC to detect forme fruste KC when tomographic and biomechanical parameters were combined [123].

#### 1.7.2. Corvis ST

The Corvis ST (CST; Oculus, Wetzlar, Germany) is a non-contact tonometer that uses a Scheimpflug camera with 4330 frames per second to record the ocular response to an air pulse in an 8 mm wide horizontal section of the cornea [124].

Several indices are calculated by this device, including Ambrósio’s Relational Thickness horizontal, biomechanically corrected IOP, stiffness parameter at first applanation, Max Inverse Radius, deformation amplitude Ratio Max, Pachy Slope, Integrated Radius, Corvis Biomechanical Index, and the Tomographic and Biomechanical Index (TBI). High diagnostic accuracy for distinguishing KC from normal eyes was only observed in some of these indices [125], and the clinical application of this tool in the diagnosis of KC is still limited. Vinciguerra et al. described the Corvis biomechanical index (CBI) [126]. In spite of affecting factors on biomechanical factors (Figure 6), CBI and TBI were designed to diagnose KC rather than detect progression; thus, the difference map is more reliable than that of Belin ABCD since it refers to the thinnest point data, which changes after CXL [127] (Table 1).

Corvis Biomechanical Factor (CBiF)—a novel measure called “E”—represents the modified linear term of the Corvis Biomechanical Index (CBI), which provides a biomechanical staging for ectasia/KC that can potentially improve the ABCD staging and detect anomalies before they are visible on tomography [128,129]. The results of cross-tabulation in recent studies showed that “E” was most equivalent to the posterior corneal curvature (“B”), whereas the anterior corneal curvature (“A”) and narrowest corneal thickness (“C”) showed a tendency toward more advanced phases [128,129].

CBI Laser Vision Correction (LVC cut of 0.2 or 0.5) has three applications: 1. CXL for post-LASIK ectasia; 2. suggestion to reoperate if there is a regression; 3. representation of the cornea or epithelium weakness.

In a study by Yang et al., DA MAX, A1T, A1V, A2V, Radius, A1DA, HCDA, A2DA, A1DLA, HCDLA, A2DLA, DLAML, A1DLAr, A1dArcL, and A2dArcL showed significant differences between KC and normal eyes [130]. The Max Inverse Radius, DA Ratio Max (2 mm), Pachy Slope, DA Ratio Max (1 mm), Integrated Radius, and CBI in KC eyes were higher than normal eyes, while the ARTh and SP-A1 were lower than those of normal eyes (all *p* < 0.05) [130].

The value of CBI is based on a logistic regression formula calculated from different Corvis ST parameters (A1V, ARTh, SP-A1, DA Ratio Max (2 mm), DA Ratio Max (1 mm), and DLA). The risk of developing ectasia is low for values < 0.25, moderate for 0.25 < CBI > 0.5, and high for CBI > 0.5 [130]. The CBI values among KC eyes after lenticule implantation showed significant change, which indicates that the value of CBI could be used to differentiate the efficacy rate.


**
AI application in Corvis ST
**


When discriminating very asymmetric ectasia and normal eyes, a random forest model with leave-one-out cross-validation (RF/LOOCV) presented the highest accuracy for TBI, BAD-D, and CBI (0.985, 0.839, and 0.822, respectively) [37]. Also, in a very asymmetric ectasia with normal topography (VAE-NT) group, an optimized TBI cut-off value of 0.29 provided 90.4% sensitivity and 96% specificity.

Also, in another recent novel study, the optimized tomographic biomechanical index of the VAE-NT group was analyzed with two different random forest algorithms (TBIv1 and TBIv2). Considering all cases, TBIv2 showed a higher AUC (0.985) than TBIv1 (0.974, *p* < 0.0001) [38].

Linear discriminant analysis and random forest were performed with the same accuracy (93%) in another study when predicting all subgroups’ severity of KC [131]. Using the variables of stiffness parameter A1, A2 time, posterior coma 0º, A2 velocity, and peak distance, a random forest model discriminated normal eyes (86%) and SKC (93%) with an accuracy rate of 89% [28].

Moreover, a support vector machine (SVM) improved AUROC to 0.948, gaining 5.7% sensitivity, 6.9% specificity, and 10.2% accuracy using Deformation Amplitude and Peak Distance at the highest concavity when classifying normal and KC eyes [132]. ANN and the finite element method were utilized in another study to evaluate the clinical data of the Pentacam and the Corvis. The accuracy of the artificial neural network in diagnosing eyes with keratoconus reached 95.5% [133].

### 1.8. Brillouin Microscopy

Brillouin microscopy looks at the interaction of laser light and spontaneous acoustic phonons within a material and can reveal mechanical qualities such as elasticity. The ex vivo Brillouin stiffness map revealed major changes in the biomechanical parameters between KC and normal corneas [134] such as corneal thinnest point changes [135].

Strain stress index (SSI) maps have been recently developed to evaluate the geographical change in biomechanical stiffness across the corneal surface in KC. The decline in fibril density and stiffness (in cone area or corneal border) might range anywhere from 0 (which indicates healthy corneas) to 60% (which indicates severe KC). It was estimated that the SSI of healthy corneas was 0.7, and a reduction from this value was achieved through an optimization process on SSI with the Corvis ST [136]. This method highlighted the dependence of corneal biomechanical behavior on the tissue microstructure, which can be an extremely useful tool for studying the pathogenesis and progression of KC disease.

## 2. Materials and Methods


**Search strategy**


According to the Preferred Reporting Items for Systematic Reviews and Meta-Analyses (PRISMA) guidelines, a comprehensive search was completed in five main electronic databases, including PubMed, Scopus, Cochrane, the Web of Science, and Embase on 27 October 2022. It did not include language or type of study restrictions. Initially, 2574 abstracts were evaluated, based on the inclusion criteria (Figure 7) and the quality assessment (Critical Appraisal Skills Program (CASP) 11-item checklist) of all included original studies, and finally 148 full-text of studies were selected for this systematic narrative review. The overall quality of the included studies was high. The following MeSH terms (keywords) were in the title of the publications used in this online search:1-(“keratoconus”);2-AND ((algorithm) OR (machine learn *) OR (deep learn *) OR (artificial intelligence) OR (automatic));3-AND ((detect *) OR (diagnos *) OR (screen *) OR (examin *) OR (analys *) OR (investigat *) OR (identif *) OR (discover *) OR (interpret *) OR (test *)).

## 3. Results

The conduct or interpretation of the index test and patient selection were clearly defined in most studies, and no applicability concerns were noted regarding these two biases. Studies showed that different AI techniques were adopted for the screening, diagnosis, and classification of KC (Figure 8).

The main corneal imaging modalities used to detect KC include topography, the Pentacam, the Corvis, SD-OCT, and the Orbscan II, with the most common data source being the Pentacam. Most of the articles included in this review used neural networks, random forests, decision trees, support vector machines, and multiple logistic regression.

Most studies analyzed local datasets since, thus far, no public KC dataset has been reported. The criteria for measuring the performance to detect KC include sensitivity, specificity, and accuracy; however, in many studies, AUC was also used.

Although the techniques showed high discrimination capacity, comparing study results is challenging because imaging modality features in small-scale studies are not recruited with biomechanical variables, post-Lasik ectasia, or corneal warpage that boost modeling cross-validation. Nevertheless, despite these obstacles, AI has tremendous potential to improve KC identification and refractive surgery screening. The scientific community’s efforts should be focused on developing platform-independent models—that can be generalized across various corneal imaging systems [31,32]—conducting external model validation on broad patient populations, stratifying the KC severity [32], and identifying and predicting the KC progression [17].

In this review, we provide a state-of-the-art, all-encompassing assessment of the indices (combination variables in addition to the value of integrated instrument characteristics) and highlight the significant shortcomings that must be addressed before any machine learning strategy can be used more effectively in early KC detection.

## 4. Discussion

DL as a machine learning (ML) approach is distinct and could progressively generate the most important features from input to output. The nascent field of machine learning, with its extraordinary potential, could play a remarkable role not only in KC detection but also in treatment modalities from contact lenses, CXL, therapeutic refractive surgery, ICRS implantation, the sizing of phakic IOLs, and the variants of keratoplasty (as described in Section 2).

Several algorithms for keratoconus identification and refractive surgery have been developed. Diagnostic systems are constructed with automated decision trees, support vector machines, and various forms of neural networks and use input from a variety of corneal imaging instruments. This review described the use of a variety of indices from the past to the future to collect more critical and useful corneal data. Although the integration of AI approaches into corneal imaging devices takes time, they are becoming more common in clinical practice. 

In general, all of the studies included in this review showed very high discrimination power between normal and KC and a lower one for SKC. However, it is difficult to directly compare the results due to the diverse definitions for the earlier stages of KC and the lack of a consistent dataset.

According to a study by Ambrósio et al., the Corvis TBI, when coupled with corneal tomography and biomechanical data, had the highest diagnostic accuracy for both KC and SKC, with greater sensitivity and specificity for KC than SKC [37]. The various cut-off points observed in these studies could be attributed to diverse patient selection criteria. The significant importance of combining these parameters together for diagnosis is suggested by the correlation of indices with the same mechanism, such as KVb and IHD (measure the corneal height) and PRFI and TBI (based on corneal tomography), in both the KC and SKC groups [89].

Clinical decision support is an important use of machine learning models [150]. The models currently in use need to be verified in a broader clinical environment since their accuracy may vary, such as when used in different areas of the world. Undoubtedly, international cooperation in conducting large-scale external evaluations of the models (which would allow for a broader understanding of the variations) would be desirable.

Machine learning offers reliable and unbiased diagnosis, which is crucial when detecting patients early, as a closer follow-up or early intervention with treatments like corneal crosslinking (CXL) could prevent disease progression, reducing the need for a corneal transplant. Machine learning techniques could benefit from using diverse sources—corneal imaging databases, clinical records, genetic data, and risk factors—to optimize their output and maximize their potential. Unsuccessful clinical machine learning models may be attributed to a lack of large patient populations to validate results; the use of diverse imaging modalities; a local participant group of various ethnic backgrounds; clinicians’ overall acceptance of machine learning techniques for diagnosis; the lack of consistent criteria for the categories of early KC, subclinical KC, and forme fruste KC [10,29,30,31,32,33,36,39,40,41,54,88,116,130]; and their relative reliability for humans.

## 5. Conclusions

Considering the challenges and novelty of the subject, this paper is the most comprehensive and all-encompassing study on keratoconus, as far as we are aware. These qualities are the result of a distinct and prominent systematic review completed with meticulous care and proficiency. We specifically made sure to concisely report the most recent and relevant data without leaving out a single subject. The novelty of this study lies in providing all the necessary information on the subject to anyone with any level of knowledge in the shortest amount of time.

The primary limitation of this study was the lack of homogeneous inputs for various studies; for instance, the staging of keratoconus patients in different studies was performed using different methods. In addition, there were fewer AI-focused investigations for some new techniques.

We concluded that more accurate KC detection requires both clinical judgment enhancement and improving the quality of the machine learning algorithm. Using AI in genetic and marker evaluation and longitudinal corneal data may predict future disease progression and identify which eyes may benefit from early intervention. 

Much research has been performed on diagnosing subclinical keratoconus using machine learning algorithms like support vector machine, I Bayes, discriminant analysis, k-nearest neighbors, random forest, decision tree, logistic regression, neural network, convolutional neural network, lasso regression, and others. When developing such approaches, it is essential to select relevant and appropriate parameter combinations from a larger parameter set and have a broad range of clinical and demographic features related to keratoconus.

## Figures and Tables

**Figure 1 diagnostics-13-02715-f001:**
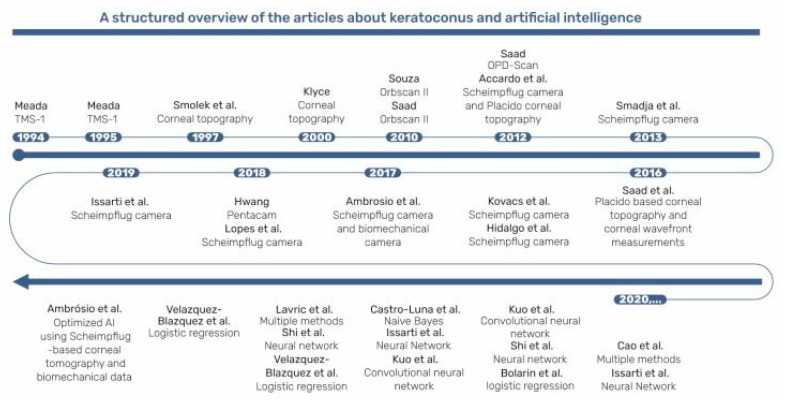
A structured overview of articles about keratoconus and artificial intelligence. Topographic Modeling System (TMS); Optical Path Difference (OPD) Scan; Ultra-High Resolution Optical Coherence Tomography (UHR-OCT) [2,10,11,21,22,23,24,25,26,27,28,29,30,31,32,33,34,35,36,37,38,39,40,41].

**Figure 2 diagnostics-13-02715-f002:**
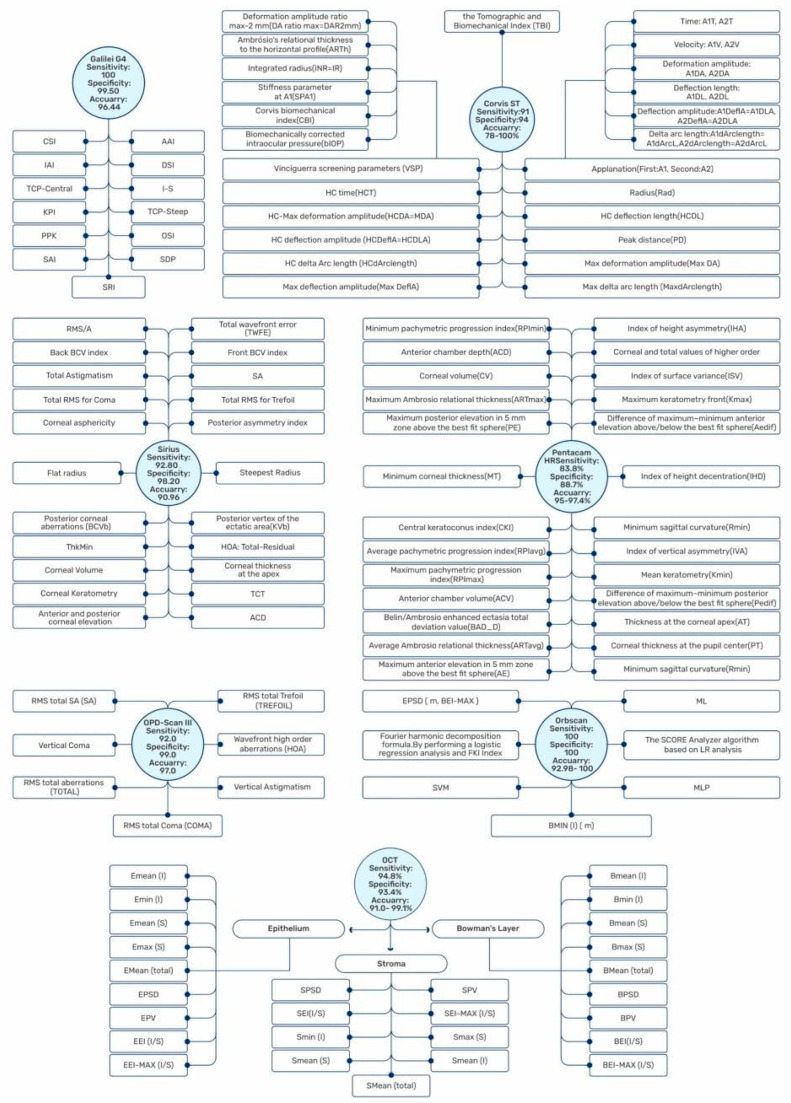
A comparison of the broad spectrum of artificial datasets about sensitivity, specificity, and accuracy represented by different machines in keratoconus. Biomechanically Corrected Intraocular Pressure (bIOP); First and Second Applanation (A1, A2); Time (T); Velocity (V); Deformation Amplitude (DA); Deflection Length (DL); Deflection Amplitude (DeflA); Delta Arc Length (dArclength); Vinciguerra Screening Parameters (VSP); Ambrósio’s Relational Thickness to the Horizontal Profile (ARTh); Integrated Radius (INR/IR); Stiffness Parameter (SP); Corvis Biomechanical Index (CBI); Radius (Rad); Highest Concavity (HC); Peak Distance (PD); Index of Height Asymmetry (IHA); Index of Surface Variance (ISV); Maximum Keratometry front (Kmax); Difference in Maximum–Minimum Anterior Elevation above/below the Best-Fit Sphere (Aedif); Index of Height Decentration (IHD); Minimal Sagittal Curvature (Rmin); Index of Vertical Asymmetry (IVA); Posterior Elevation (PE); Minimum Keratometry (Kmin); Maximum Posterior Elevation in 5 mm Zone above the Best-Fit Sphere (PE); Difference in Maximum–Minimum Posterior Elevation above/below the Best-Fit Sphere (Pedif); Maximum Anterior Elevation in 5 mm Zone above the Best-Fit Sphere (AE); Thickness at the Corneal Apex (AT); Corneal Thickness at the Pupil Center (PT); Minimum Sagittal Curvature (Rmin); Minimum Pachymetric Progression Index (RPImin); Average Pachymetric Progression Index (RPIavg); Maximum Pachymetric Progression Index (RPImax); Anterior Chamber Depth (ACD); Corneal Volume (CV); Maximum Ambrósio Relational Thickness (ARTmax); Average Ambrósio Relational Thickness (ARTavg); Minimum Corneal Thickness (MT); Central KC Index (CKI); Anterior Chamber Volume (ACV); Belin–Ambrósio Enhanced Ectasia Total Deviation value (BAD_D); Surface Regularity Index (SRI); Standard Deviation of Corneal Power (SDP); Opposite Sector Index (OSI); Surface Asymmetry Index (SAI); Percentage Probability of KC (PPK); KC Prediction Index (KPI); Asphericity Asymmetry Index (AAI); Differential Sector Index (DSI); Inferior–Superior (I–S) Index; Total Corneal Power (TCP); Center/Surround Index (CSI); Irregular Astigmatism Index (IAI); Root Mean Square (RMS); Baiocchi Calossi Versaci (BCV); Posterior Corneal Aberrations (BCVb); Thinnest Point of the Cornea (ThkMin); Total Wavefront Error (TWFE); Spherical Aberrations (SA); KC Vertex back (KVb); Higher Order Aberration (HOA); Thinnest Corneal Thickness (TCT); Machine Learning (ML); Multilayer Perceptron (MLP); Screening Corneal Objective Risk of Ectasia (SCORE); Support Vector Machine (SVM); Logistic Regression (LR); Fourier-Incorporated KC Detection Index (FKI); Standard Deviation of Thickness Profile between Individual and Normal Patterns of Epithelium, Bowman’s Layer, and Stroma (EPSD, BPSD, SPSD); Profile Variation in epithelium, Bowman’s Layer, or Stroma Thickness Profile within Each Individual (EPV, BPV, SPV); Ectasia Index of Epithelium, Bowman’s Layer, or Stroma (EEI, BEI, SEI); Maximum Ectasia Index of Epithelium Layer, Bowman’s Layer, or Stroma (EEI-MAX, BEI-MAX, SEI-MAX); Mean Thickness of Epithelium, Bowman’s Layer, or Stroma (EMean, BMean, SMean); Thinnest Thickness of the Inferior Epithelium, Bowman’s Layer, or Stroma Thickness Map (Emin, Bmin, Smin); Thickest Thickness of the Superior Epithelium, Bowman’s Layer, or Stroma Thickness Map (Emax, Bmax, Smax).

**Figure 3 diagnostics-13-02715-f003:**
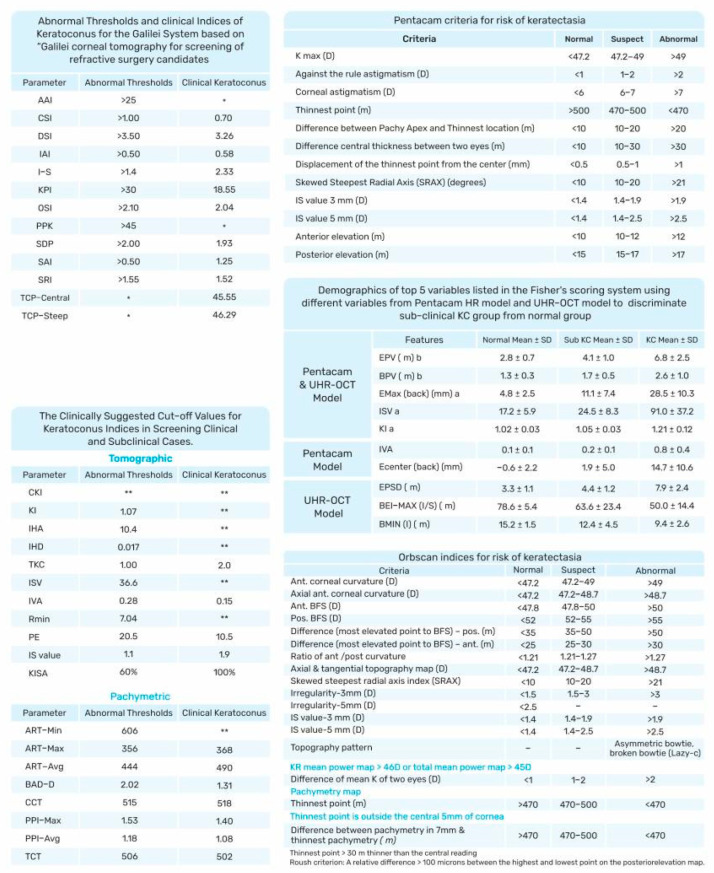
Abnormal and suggestive of KC thresholds for different devices. Best-Fit Sphere (BFS); Asphericity Asymmetry Index (AAI); Center/Surround Index (CSI); Differential Sector Index (DSI); Irregular Astigmatism Index (IAI); Inferior–Superior (I–S) Index; KC Prediction Index (KPI); Opposite Sector Index (OSI); Percentage Probability of KC (PPK); Standard Deviation of Corneal Power (SDP); Surface Asymmetry Index (SAI), Surface Regularity Index (SRI); Total Corneal Power (TCP); Central KC Index (CKI); KC Index (KI); Index of Height Asymmetry (IHA); Index of Height Decentration (IHD); Pentacam Topographical KC Classification (TKC); Index of Surface Variance (ISV); Index of Vertical Asymmetry (IVA); Minimal Sagittal Curvature (Rmin); Posterior Elevation (PE); Ambrósio’s Relational Thickness (ART); Belin–Ambrósio Enhanced Ectasia Display Total Deviation (BAD_D) Value; Central Corneal Thickness (CCT); Pachymetric Progression Indices (PPI); Thinnest Corneal Thickness (TCT); Maximum Keratometry (Kmax); Relative Skewing of the Steepest Radial Axes (SRAX); Epithelium Profile Variation (EPV); Bowman’s Layer Profile Variation (BPV); Maximum Elevation (Emax); Central Elevation (Ecenter); Epithelium Profile Standard Deviation (EPSD); Maximum Ectasia Index of Bowman’s Layer (BEI-MAX); Thinnest Thickness of the Inferior Bowman’s Layer Thickness Map (Bmin); Best-Fit Sphere (BFS). * and **: Although several studies have discussed these parameters, there is no consequence on the thresholds and cutoff values.

**Figure 4 diagnostics-13-02715-f004:**
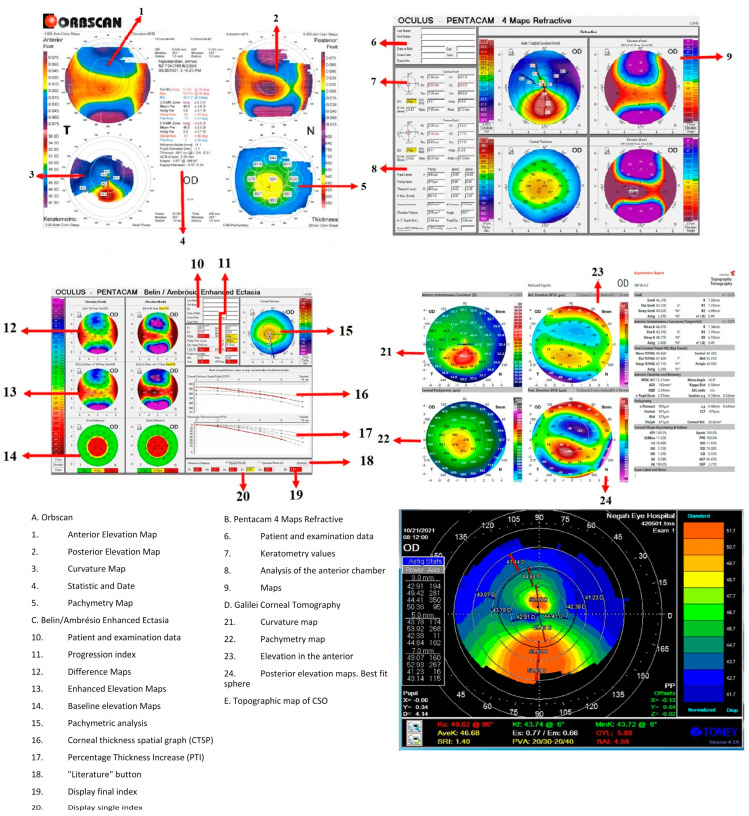
The important feature of four devices in a keratoconus eye from one patient. The Orbscan II, the Pentacam AXL/wave (four maps refractive display), Belin/Ambrosio enhanced ectasia display with a pachymetric map, the Galilei G4, and axial map Sirius the lowest part).

**Figure 5 diagnostics-13-02715-f005:**
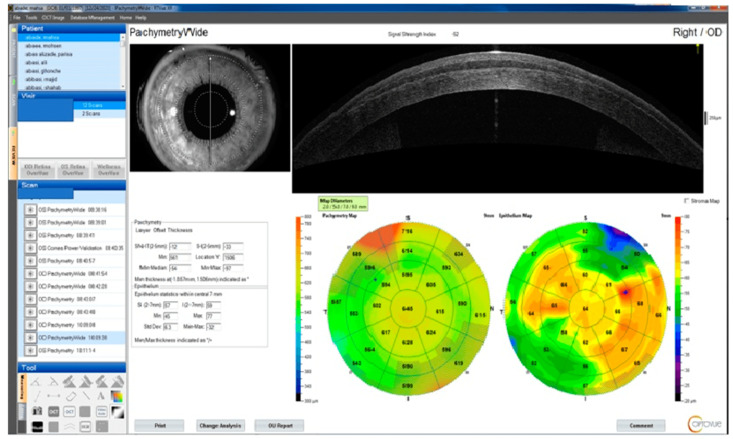
Comparison of epithelial thickness measurements, RTVue SD-OCT device (Optovue, Inc., Fremont, CA, USA), and epithelial thickness map of a patient after Smile’s derived lenticule implantation.

**Figure 6 diagnostics-13-02715-f006:**
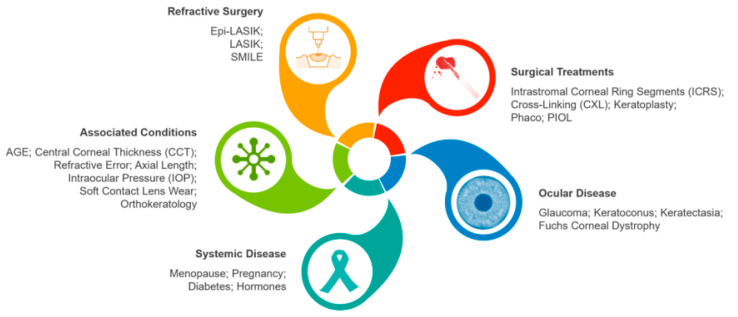
Factors affecting corneal biomechanical properties. Laser-Assisted In Situ Keratomileusis (LASIK); Epithelial LASIK (Epi-LASIK); Small Incision Lenticule Extraction (SMILE); IntraCorneal Ring Segments (ICRS); Corneal Collagen Crosslinking (CXL); Phakic Intraocular Lens (pIOL); Central Corneal Thickness (CCT); Intraocular Pressure (IOP).

**Figure 7 diagnostics-13-02715-f007:**
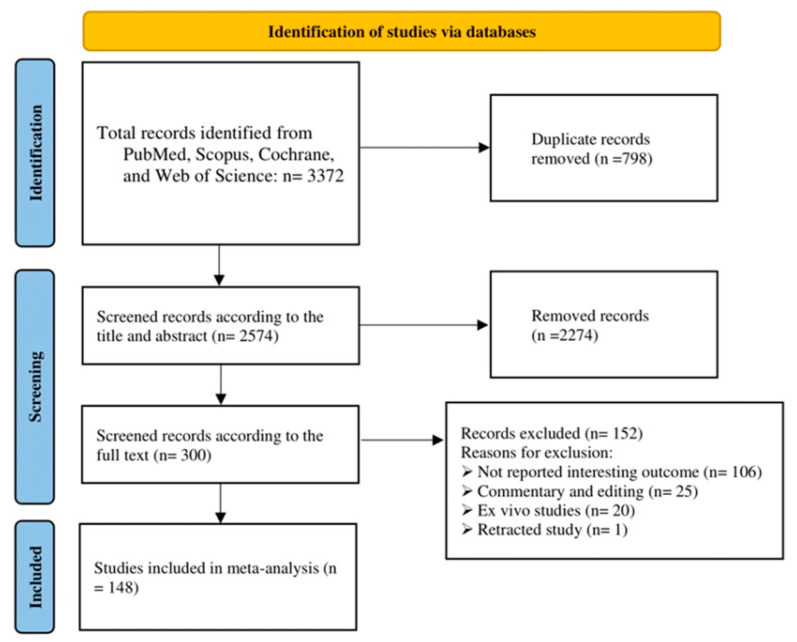
Study selection according to search strategy.

**Figure 8 diagnostics-13-02715-f008:**
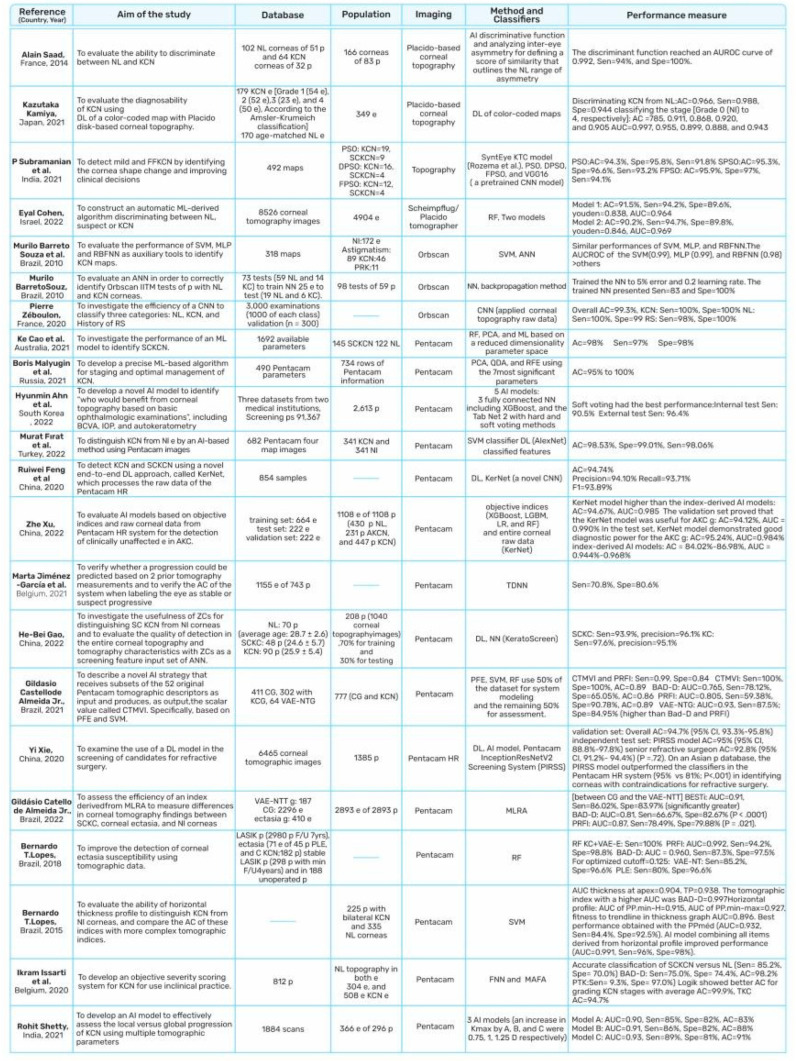
Summary of automatic screening, diagnosis, and classification methods for KC. Patients (p); Eyes (e); Group (g); KC group (KCG); control group (CG); Normal (Nl); Follow up (F/U); Keratoconus (KC); Clinical KC (CKC); Subclinical KC (SKC); Advanced KC (AKC); very asymmetric ectasia (VAE); Very Asymmetric Ectasia but with Normal Corneal Topography (VAE-NTG); Ocular Surface Disorders (OSD); Epithelial Basement Membrane Dystrophy (EBMD); Dry Eye Disease (DED); Keratitis Precipitate (KP), Subepithelial Opacity (SEO); Area Under the Curve (AUC); Area Under the Receiver Operator Characteristic Curve (AUROCC); Accuracy (AC); Sensitivity (Sen); Specificity (Spe); Recall (R); Purity (Pu); Belin–Ambrósio Deviation Index (BAD-D); Corneal Tomography Multivariate Index (CTMVI); Pentacam Topographical KC Classification (TKC); Tomographic–Biomechanical Parameter (TBI); Zernike Coefficients (ZC); Corneal Epithelial Thickness (ET); Deformation Amplitude (DA); Peak Distance (PD) at the Highest Concavity; Boosted Ectasia Susceptibility Tomography Index (BESTi); Multiple Logistic Regression Analysis (MLRA); Artificial intelligence (AI); Paraconsistent Feature Engineering (PFE); Support Vector Machine (SVM); Pentacam Random Forest index (PRFI); Artificial Neural Network (ANN); Flower Pollination Algorithm (FPA); Random Forest (RF); Radial Basis Function NN (RBFNN); Particle Swarm Optimization (PSO); Fractional Order PSO (FPSO); Discrete PSO (DPSO); Linear Regression (LR); Time Delay Neural Network (TDNN); Convolutional NN (CNN); Feedforward Neural Network (FNN); Multilayer Perceptron (MLP); Local Binary Pattern (LBP); Local Directional Pattern (LDP); Local Optimal Oriented Pattern (LOOP); Cat Swarm Optimization (CSO); Linear Discriminant Analysis (LDA); Principal Component Analysis (PCA); Quadratic Discriminant Analysis (QDA); Masked Face Analysis (MAFA); Anterior Segment Optical Coherence Tomography (AS-OCT); Photorefractive Keratectomy (PRK); Phototherapeutic Keratectomy (PTK); Penetrating Keratoplasty (PK); Lamellar Keratoplasty (LK); Laser-Assisted In Situ Keratomileusis (LASIK); Post-LASIK Ectasia (PLE). [17,20,25,26,27,28,29,32,37,41,47,48,49,56,75,86,91,95,118,119,130,131,132,137,138,139,140,141,142,143,144,145,146,147,148,149].

**Table 1 diagnostics-13-02715-t001:** Parameters of Corvis ST.

Corvis ST Parameters		
Parameters	Abbreviation	Description
Biomechanically corrected intraocular pressure	bIOP	Derived by finite element simulations that take into account the influence of central corneal thickness, age, and dynamic corneal response (DCR) parameters
First applanation	A1	Moment at the first applanation of the cornea during the air puff
A1 time (ms)	A1T(T1)	Time from start to A1
A1 velocity (m/s)	A1V(V1)	Velocity (speed) of corneal apex at A1
A1 deformation amplitude	A1DA	Moving distance of the corneal apex from the initial position to that at the A1 time
A1 deflection length	A1DL	Length of the flattened cornea at A1
A1 deflection amplitude	A1DeflA, A1DLA	After approaching the highest displacement secondary to WEM, the whole eye displays a nonlinear motion in the ant–post direction, so A1DeflA is similar to A1DA without WEM
A1 delta arc length	A1dArclength, A1dArcL	Change in arc length from the initial state to A1, in a defined 7 mm zone
Second applanation	A2	Moment at the first applanation of the cornea during the air puff
A2 time (ms)	A2T(T2)	Time from start to A2
A2 velocity (m/s)	A2V	Speed of corneal apex at A2
A2 deformation amplitude	A2DA	Moving distance of the corneal apex from the initial position to that at A2 time
A2 deflection length	A2DL	Length of the flattened cornea at A2
A2 deflection amplitude	A2DeflA, A2DLA	Similar to A2DA without whole eye movement
A2 delta arc length	A2dArclength, A2dArcL	Change in arc length from the initial state to A2, in a defined 7 mm zone
Highest (maximum) concavity	HC, MC	Moment that the cornea assumes its maximum concavity during the air puff
HC time	HCT	Time to reach the maximum deformation
Radius (mm)	Rad	Central curvature radius at the HC state secondary to parabolic fit
HC (Max) deformation amplitude	HCDA, MDA	Maximum depth of ant–post corneal displacement at the moment of maximum concavity
HC deflection length	HCDL	Length of the flattened cornea at highest concavity
HC deflection amplitude	HCDeflA, HCDLA	“Displaced” area of the cornea in the horizontal plane secondary to corneal deformation
Peak distance	PD	Distance between the two peaks of the cornea in temporal–nasal direction at the maximum concavity state, which is not the same as the deflection length
HC delta Arc length	HCdArclength	Change in arc length in a defined 7 mm zone during HC from the initial state
Maximum	Max	Similar to HC
Max deformation amplitude	Max DA	Distance of the corneal apex movement from the initiation of the deformation to the HC
Max deflection amplitude	Max DeflA	Ratio between the deformation/deflection amplitude at the apex and the average deformation/deflection amplitude in a nasal and temporal zone 1 or 2 mm (2 mm for DefA ratio) from the center; higher values (greater 1) of DA Ratio and DefA Ratio can be associated with less resistant corneas
Max delta arc length	MaxdArclength	Change in arc length during the highest concavity moment from the initial state, in a 7 mm with horizontal direction (3.5 mm from the apex to both sides)
Vinciguerra screening parameters (VSP)		
Deformation amplitude ratio max (2 mm)	DA ratio max(DAR2mm)	Ratio between the deformation amplitude at the apex and the average deformation amplitude measured at 2 mm central–peripheral
Ambrósio’s relational thickness to the horizontal profile	ARTh	Ratio between the deformation amplitude at the apex and the average deformation amplitude measured at 2 mm from the center
Integrated radius	INR(IR)	Area under the inverse concave radius vs. time curve; in fact, 1/R is plotted during the time of an air pulse and is entirely measured between the period of first and second applanations
Stiffness parameter at A1	SP-A1	Corneal stiffness at A1, the ratio of resultant pressure to deflection amplitude
Corvis biomechanical index	CBI	Overall biomechanical index for keratoconus detection

Dynamic corneal response = DCR; Inverse Concave Radius = (1/R); Whole Eye globe Movement = WEM; Anterior–Posterior = Ant–Post.

## Data Availability

According to Preferred Reporting Items for Systematic Reviews and Meta-Analyses (PRISMA) guidelines, a comprehensive search was completed using five primary electronic public databases, including PubMed, Scopus, Cochrane, the Web of Science, and Embase.

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
