# Peer review of "Keratoconus Diagnosis: From Fundamentals to Artificial Intelligence: A Systematic Narrative Review"

_diagnostics, 2023, doi:10.3390/diagnostics13162715_

Round 1

Reviewer 1 Report

The manuscript entitled "Keratoconus diagnosis: from fundamentals to artificial intelligence. A Systematic Narrative Review".

Observation

Abstract

Please explain the abbreviation KC and SK. 

Figure 1 and 2 are not visible. Same observation for the next figures.

Please provide at the end of Introduction the aim of the study - the hypothesis you wanted to be tested - not as a Conclusion. The information at the end of Introduction sounds like a Conclusion.

You also need to mention the Material and Methods (what kind of review is this manuscript and how you selected the paper for this study).

All the figures has to be reorganized because it is very difficult to to be read. 

At the end of the study, please write the study limitation.

Conclusion - what is the novelty of this study? Why you consider is useful for medical practice?

Author Response

The manuscript entitled "Keratoconus diagnosis: from fundamentals to artificial intelligence. A Systematic Narrative Review".

Observation

Abstract

Please explain the abbreviation KC and SK.

Dear Reviewer 1,

Thank you for pointing that out. The abbreviation of Subclinical Keratoconus, which was falsely written as SK instead of SKC was replaced with its long form, along with KC which Keratoconus replaces.

Figure 1 and 2 are not visible. Same observation for the next figures.

We appreciate your comments very much. Figure 3 has been updated. Also, We have enhanced the figures’ resolution as much as possible. However, if the journal’s editorial team has any option that could enhance them further, we'd gladly send the original files and pay for the service. Also, if there is not a strict limitation on the page count, having bigger tables on multiple pages would surely improve the reading experience and help appreciate the effort put into them.

Please provide at the end of Introduction the aim of the study - the hypothesis you wanted to be tested - not as a Conclusion. The information at the end of Introduction sounds like a Conclusion.

Thanks for providing your insight. Based on your valuable comment the Introduction has been revised.

You also need to mention the Material and Methods (what kind of review is this manuscript and how you selected the paper for this study).

Your comments are of great value to us. The type of the manuscript is now mentioned in the Materials and Methods. However, figure 7, which extensively outlines the study selection process, should be enough to alleviate your second concern.

Figure 7. Study selection according to search strategy.

All the figures has to be reorganized because it is very difficult to to be read.

Thank you for your insightful comments. Figure 3 has been updated. Also, We have enhanced the figures’ resolution as much as possible. However, if the journal’s editorial team has any option that could enhance them further, we'd gladly send the original files and pay for the service. Also, if there is not a strict limitation on the page count, having bigger tables on multiple pages would surely improve the reading experience and help appreciate the effort put into them.

At the end of the study, please write the study limitation.

Thanks for pointing that out. The study limitation has been added to lines 768-771.

Conclusion - what is the novelty of this study? Why you consider is useful for medical practice?

Thank you for your time and effort. The Conclusion section has been expanded.

“Considering the challenges and novelty of the subject, this paper is the most comprehensive and all-encompassing study on keratoconus, as far as we are aware. These qualities are the result of a distinct and prominent systematic review done with meticulous care and proficiency. We specifically made sure to report the most recent and relevant data concisely without leaving out a single subject. The novelty of this study lies in providing all the necessary information on the subject to anyone with any level of knowledge in the shortest amount of time.”

Also, based on your suggestion, we have moved a part of the introduction to the conclusion.

Reviewer 2 Report

Keratoconus diagnosis: from fundamentals to artificial intelligence. A Systematic Narrative Review

Line 18. It reads:

 Abstract: The remarkable recent advances in managing keratoconus as the most common corneal ectasia encourage researchers to do more studies about the disease. Despite the abundance of information about keratoconus, there are still debates about detecting mild cases. At the same time, early detection allows us to carry out the most noninvasive treatments to achieve an emmetropic eye. In the words of Confucius, “Study the past if you would define the future,” this review has captured more critical and valuable corneal data from fundamental to artificial intelligence in keratoconus patients. Diagnostic systems are built using automated decision trees, support vector machines, and various types of neural networks, with input from various corneal imaging equipment. Although it takes time for artificial intelligence techniques to be integrated into corneal imaging devices, they are becoming more popular in clinical practice. Most of the studies in this review demonstrated very high discrimination power between normal and KC and a lower one for SK.

Comment: The statement “… the most noninvasive treatments to achieve an emmetropic eye…” is not really realistic in keratoconus. Therefore, it should be deleted. Any available treatment for keratoconus, i.e. intracorneal ring segments or crosslinking, are not completely noninvasive.   The Confucius' wisdom, “Study the past if you would define the future” does not really fit in the context.  The adjective “vital” does not really fit the context.

Consider modifying to:

"Abstract: The remarkable recent advances in managing keratoconus, the most common corneal ectasia, have encouraged researchers to conduct further studies on the disease. Despite the abundance of information about keratoconus, debates persist regarding the detection of mild cases. Early detection plays a crucial role in facilitating less invasive treatments. This review encompasses corneal data ranging from basic sciences to the application of artificial intelligence in keratoconus patients. Diagnostic systems utilize automated decision trees, support vector machines, and various types of neural networks, incorporating input from various corneal imaging equipment. Although the integration of artificial intelligence techniques into corneal imaging devices may take time, their popularity in clinical practice is increasing. Most studies reviewed herein demonstrated a high discriminatory power between normal and keratoconus cases, with a relatively lower discriminatory power for subclinical keratoconus."

Line 34. It reads: “Keratoconus (KC) is a progressive corneal ectasia with a prevalence that varies from 1 in 50 in 35 Central India to 1 in 2000 in the United States [2, 3].”

Comment: It could be useful for the reader to make a short mention to etiology at this moment.

Consider modifying to: “Keratoconus (KC) is a progressive corneal ectasia characterized by a thinning and protrusion of the cornea. The development of KC is influenced by both genetic and environmental factors, with environmental factors such as eye rubbing and nocturnal ocular compression appearing to play a more significant role. The prevalence of KC varies across different regions, ranging from 1 in 50 individuals in Central India to 1 in 2000 individuals in the United States [2, 3]."

Additional references:

1-Gatinel D, Galvis V, Tello A, et al. Obstructive Sleep Apnea-Hypopnea Syndrome and Keratoconus: An Epiphenomenon Related to Sleep Position?. Cornea. 2020;39(4):e11-e12. doi:10.1097/ICO.0000000000002219

2-Gurnani B, Kaur K. Evolving concepts in etiopathogenesis of keratoconus: Is it quasi-inflammatory or inflammatory?. Indian J Ophthalmol. 2023;71(6):2609-2610. doi:10.4103/ijo.IJO_783_22

3-Seth I, Bulloch G, Vine M, et al. The association between keratoconus and allergic eye diseases: A systematic review and meta-analysis. Clin Exp Ophthalmol. 2023;51(4):O1-O16. doi:10.1111/ceo.14215

Line 36: It reads: “Patients with KC are the second-largest group requiring corneal transplantation worldwide, and young adults and children are the most affected by the condition [2, 4].”

Comment:

The indications for corneal transplantation have changed during the last decades, and keratoconus is now a much less frequent reason for the surgery in many countries. Therefore, consider modifying to:

“In spite of therapeutic advances, including corneal collagen crosslinking and intracorneal ring segments, patients with KC are still an important group requiring corneal transplantation worldwide, and young adults and children are the most affected by the condition [2, 4].”

Additional references:

1-Deshmukh R, Ong ZZ, Rampat R, et al. Management of keratoconus: an updated review. Front Med (Lausanne). 2023;10:1212314. Published 2023 Jun 20. doi:10.3389/fmed.2023.1212314

2-Galvis V, Tello A, Laiton AN, Salcedo SLL. Indications and techniques of corneal transplantation in a referral center in Colombia, South America (2012-2016). Int Ophthalmol. 2019;39(8):1723-1733. doi:10.1007/s10792-018-0994-z.

Line 42. It reads: “Several studies described multiple methods in KC detection (Figure 1), because genetic [8-11], atopic [12], environmental, and mechanical [8] factors may be involved in KC development and progression [13], resulting in a complexity that may benefit from an artificial intelligence (AI) approach, which has already been used to forecast keratoconus progression [14-17].”

Comment:

Consider modifying to: “Various methods have been described for detecting keratoconus (KC) (Figure 1), primarily utilizing corneal topographers or tomographers. However, as mentioned, KC is a multifactorial condition involving genetic factors [8-11], and also environmental factors such as atopy [12], and repetitive mechanical corneal trauma [8] in its development and progression [13]. Due to the complex nature of KC, there is potential benefit in utilizing artificial intelligence (AI) approaches, including also corneal biomechanical information, which have already shown promise in forecasting the progression of keratoconus [14-17].”

Author Response

Line 18. It reads:

Abstract: The remarkable recent advances in managing keratoconus as the most common corneal ectasia encourage researchers to do more studies about the disease. Despite the abundance of information about keratoconus, there are still debates about detecting mild cases. At the same time, early detection allows us to carry out the most noninvasive treatments to achieve an emmetropic eye. In the words of Confucius, “Study the past if you would define the future,” this review has captured more critical and valuable corneal data from fundamental to artificial intelligence in keratoconus patients. Diagnostic systems are built using automated decision trees, support vector machines, and various types of neural networks, with input from various corneal imaging equipment. Although it takes time for artificial intelligence techniques to be integrated into corneal imaging devices, they are becoming more popular in clinical practice. Most of the studies in this review demonstrated very high discrimination power between normal and KC and a lower one for SK.

Comment: The statement “… the most noninvasive treatments to achieve an emmetropic eye…” is not really realistic in keratoconus. Therefore, it should be deleted. Any available treatment for keratoconus, i.e. intracorneal ring segments or crosslinking, are not completely noninvasive.   The Confucius' wisdom, “Study the past if you would define the future” does not really fit in the context.  The adjective “vital” does not really fit the context.

Consider modifying to:

"Abstract: The remarkable recent advances in managing keratoconus, the most common corneal ectasia, have encouraged researchers to conduct further studies on the disease. Despite the abundance of information about keratoconus, debates persist regarding the detection of mild cases. Early detection plays a crucial role in facilitating less invasive treatments. This review encompasses corneal data ranging from basic sciences to the application of artificial intelligence in keratoconus patients. Diagnostic systems utilize automated decision trees, support vector machines, and various types of neural networks, incorporating input from various corneal imaging equipment. Although the integration of artificial intelligence techniques into corneal imaging devices may take time, their popularity in clinical practice is increasing. Most studies reviewed herein demonstrated a high discriminatory power between normal and keratoconus cases, with a relatively lower discriminatory power for subclinical keratoconus."

Dear Reviewer 1,

Thank you very much for your keen observation and valuable insight. The abstract has been modified per your instructions. (Lines 18-28)

Line 34. It reads: “Keratoconus (KC) is a progressive corneal ectasia with a prevalence that varies from 1 in 50 in 35 Central India to 1 in 2000 in the United States [2, 3].”

Comment: It could be useful for the reader to make a short mention to etiology at this moment.

Consider modifying to: “Keratoconus (KC) is a progressive corneal ectasia characterized by a thinning and protrusion of the cornea. The development of KC is influenced by both genetic and environmental factors, with environmental factors such as eye rubbing and nocturnal ocular compression appearing to play a more significant role. The prevalence of KC varies across different regions, ranging from 1 in 50 individuals in Central India to 1 in 2000 individuals in the United States [2, 3]."

Additional references:

1-Gatinel D, Galvis V, Tello A, et al. Obstructive Sleep Apnea-Hypopnea Syndrome and Keratoconus: An Epiphenomenon Related to Sleep Position?. Cornea. 2020;39(4):e11-e12. doi:10.1097/ICO.0000000000002219

2-Gurnani B, Kaur K. Evolving concepts in etiopathogenesis of keratoconus: Is it quasi-inflammatory or inflammatory?. Indian J Ophthalmol. 2023;71(6):2609-2610. doi:10.4103/ijo.IJO_783_22

3-Seth I, Bulloch G, Vine M, et al. The association between keratoconus and allergic eye diseases: A systematic review and meta-analysis. Clin Exp Ophthalmol. 2023;51(4):O1-O16. doi:10.1111/ceo.14215

We appreciate the time and effort you have invested in helping us improve the manuscript. The text has been duly modified and your suggested references have also been included. (Lines 44-46)

Line 36: It reads: “Patients with KC are the second-largest group requiring corneal transplantation worldwide, and young adults and children are the most affected by the condition [2, 4].”

Comment:

The indications for corneal transplantation have changed during the last decades, and keratoconus is now a much less frequent reason for the surgery in many countries. Therefore, consider modifying to:

“In spite of therapeutic advances, including corneal collagen crosslinking and intracorneal ring segments, patients with KC are still an important group requiring corneal transplantation worldwide, and young adults and children are the most affected by the condition [2, 4].”

Additional references:

1-Deshmukh R, Ong ZZ, Rampat R, et al. Management of keratoconus: an updated review. Frontiers in medicine. 2023;10:1212314. Published 2023 Jun 20. doi:10.3389/fmed.2023.1212314

2-Galvis V, Tello A, Laiton AN, Salcedo SLL. Indications and techniques of corneal transplantation in a referral center in Colombia, South America (2012-2016). International ophthalmology. 2019;39(8):1723-1733. doi:10.1007/s10792-018-0994-z.

 Thank you for pointing it out. The text has been duly modified and your suggested references have also been included. (Lines 48-51)

Line 42. It reads: “Several studies described multiple methods in KC detection (Figure 1), because genetic [8-11], atopic [12], environmental, and mechanical [8] factors may be involved in KC development and progression [13], resulting in a complexity that may benefit from an artificial intelligence (AI) approach, which has already been used to forecast keratoconus progression [14-17].”

Comment:

Consider modifying to: “Various methods have been described for detecting keratoconus (KC) (Figure 1), primarily utilizing corneal topographers or tomographers. However, as mentioned, KC is a multifactorial condition involving genetic factors [8-11], and also environmental factors such as atopy [12], and repetitive mechanical corneal trauma [8] in its development and progression [13]. Due to the complex nature of KC, there is potential benefit in utilizing artificial intelligence (AI) approaches, including also corneal biomechanical information, which have already shown promise in forecasting the progression of keratoconus [14-17].”

Thank you for your insightful comments. The text has been modified per your instructions. (Lines 55-61)

Round 2

Reviewer 1 Report

Congratulation for your paper!